# Evaluation of Natural and Factitious Food Sources for *Pronematus ubiquitus* on Tomato Plants

**DOI:** 10.3390/insects12121111

**Published:** 2021-12-13

**Authors:** Marcus V. A. Duarte, Dominiek Vangansbeke, Juliette Pijnakker, Rob Moerkens, Alfredo Benavente, Yves Arijs, Ana Lizbeth Flores Saucedo, Felix Wäckers

**Affiliations:** 1Biobest Group N.V., Isle Velden 18, 2260 Westerlo, Belgium; Dominiek.vangansbeke@biobestgroup.com (D.V.); juliette.pijnakker@biobestgroup.com (J.P.); Rob.Moerkens@biobestgroup.com (R.M.); Alfredo.Benavente@biobestgroup.com (A.B.); Yves.Arijs@biobestgroup.com (Y.A.); Felix.Wackers@biobestgroup.com (F.W.); 2Department of Molecular Phytopathology and Mycotoxin Research, Georg-August-Universität Göttingen, 37073 Göttingen, Germany; anaflores.0591@gmail.com

**Keywords:** biological control, alternative food, Tydeoidea

## Abstract

**Simple Summary:**

Biocontrol practitioners have increasingly released generalist predators to control a variety of greenhouse pests. In this study, the effects of alternative food sources on the oviposition rate and the population dynamics of the mite *Pronematus ubiquitus* were assessed. The alternative food sources were selected among those the mite may encounter or be applied to the tomato crop.

**Abstract:**

*Pronematus ubiquitus* (McGregor) is a small iolinid mite that is capable of establishing on tomato plants. Once established, this mite has been shown to control both tomato russet mite, *Aculops lycopersici* (Tryon) (Acari: Eriophyidae), and tomato powdery mildew (*Oidium neolycopersici* L. Kiss). In the present study, we explored the effects of a number of alternative food sources on the oviposition rate in the laboratory. First, we assessed the reproduction on food sources that *P. ubiquitus* can encounter on a tomato crop: tomato pollen and powdery mildew, along with tomato leaf and *Typha angustifolia* L. In a second laboratory experiment, we evaluated the oviposition rate on two prey mites: the astigmatid *Carpoglyphus lactis* L. (Acari: Carpoglyphidae) and the tarsonemid *Tarsonemus fusarii* Cooreman (Acari: Tarsonemidae). Powdery mildew and *C. lactis* did not support reproduction, whereas tomato pollen and *T. fusarii* did promote egg laying. However, *T. angustifolia* pollen resulted in a higher oviposition in both experiments. In a greenhouse trial on individual caged tomato plants, we evaluated the impact of pollen supplementation frequency on the establishment of *P. ubiquitus*. Here, a pollen addition frequency of every other week was required to allow populations of *P. ubiquitus* to establish.

## 1. Introduction

In contrast to the long-held conviction that specialist natural enemies are required to guarantee effective pest control [1], biocontrol practitioners have been increasingly releasing generalist predators to control a several of greenhouse pests [2,3]. Generalist predators possess several interesting properties, which explain this shift: (1) Generalist predators feed on several pest species. A single generalist predator species thus simultaneously affects populations of multiple pest species [4,5]. (2) They are more likely to be able to exploit alternative food sources, such as pollen, fungi, or plant sap [6,7], which may enable them to build up populations in the absence of prey food [8,9,10,11,12]. (3) Generalist predators are generally easier and cheaper to mass-produce [13]. For example, phytoseiid predatory mites can be mass-produced on factitious astigmatid prey mites [14,15]. As a result, generalist phytoseiid predatory mites, such as *Amblyseius swirskii* (Athias-Henriot), *Neoseiulus californicus* (McGregor), and *Neoseiulus cucumeris* (Oudemans), are now among the most commercialized arthropod biocontrol agents worldwide [16,17], and have been introduced in a wide range of crops, mostly in protected cultivation. In protected crops such as pepper and cucumber, phytoseiids successfully control thrips, whiteflies, and spider mites [18,19,20,21,22,23]. Unlike on cucumber and pepper, phytoseiid predatory mites have difficulties building up populations on tomato crops, even when having access to prey food shown to be suitable in laboratory trials, such as the tomato russet mite (TRM), *Aculops lycopersici* (Tryon) (Acari: Eriophyidae), and several species of whiteflies [24,25,26]. The inability of Phytoseiidae to establish on tomato plants is attributed to the presence of glandular trichomes, which hamper their movement, especially those found in the young leaves [27,28]. Unlike the larger phytoseiid predators, smaller mites from the family Iolinidae (superfamily Tydeoidea) are not affected by the presence of glandular trichomes and can successfully establish on tomato plants [29,30]. For example, *Pronematus ubiquitus* (McGregor) is a small (<300 µm) omnivorous mite that feeds on a variety of plant-derived food sources, such as pollen and plant sap [3,31]. This predatory mite also feeds on small prey food, such as Eriophyid [32] and Tetranychid mites [32,33]. In addition to plant-provided food sources and prey, these mites are also known to feed on fungi [34,35]. Recently, this predatory mite was found to effectively control two key problems on tomato crops: TRM, *A. lycopersici,* and powdery mildew (*Oidium neolycopersici* L. Kiss) [3]. This predatory mite can be pre-established and build up large populations by supplementing tomato plants with *Typha* sp. pollen [3,31].

The nutritional value of food sources can influence the biological parameters of an organism. When on the tomato crop, *P. ubiquitus* can encounter food sources such as tomato pollen and conidia of powdery mildew [3]. Other Iolinidae predators have also been reported to feed on fungi [36,37]. However, the influence of a specific food source on the biological parameters of a mite species can also be affected by its origin. Therefore, this study aimed to select the food supplement that can optimize the reproduction of one or two lines of *P. ubiquitus*, collected in Belgium and The Netherlands.

Several food sources have been tested to support the establishment of generalist predators in greenhouse crops [23,38,39], including pollen [8,40,41,42], frozen eggs of *Ephestia kuehniella* Zelle [23], and *Artemia* sp. cysts [12,43]. Here, we explored the potential of two species of prey mites for supporting a population of *P. ubiquitus* on a tomato crop, namely the dried-fruit mite *Carpoglyphus lactis* (L.) (Acari: Carpoglyphidae) and *Tarsonemus fusarii* Cooreman (Acari: Tarsonemidae). The former is a conventionally used astigmatid prey mite that can support populations of phytoseiid predatory mites on crops [44,45,46,47]. In addition, several Tarsonemidae species were found to be suitable prey mites for phytoseiids [48,49,50]. Finally, based on the outcome of the feeding experiments in the laboratory, a greenhouse trial was conducted to assess the optimal application frequency of *Typha* pollen for the establishment of *P. ubiquitus* on tomato plants. 

## 2. Materials and Methods

### 2.1. Mites and Plants

Two lines of *P. ubiquitus* were collected in Belgium (Merelbeke) and The Netherlands (Venlo), from blackberry (*Rubus* sp.) and grape (*Vitis* sp.) plants, respectively. These lines were mass-reared in the production facilities of Biobest N.V. on a diet consisting of *T. angustifolia* pollen (Nutrimite^TM^). *Tarsonemus fusarii* was produced on *Aspergillus oryzae* fungus grown on rice flakes (De Halm B.V.) as described by Vangansbeke et al. [49]. A mixture of yeast flakes and bran was used to produce *C. lactis*. Both *T. fusarii* and *C. lactis* were produced in climate chambers at 22 ± 1 °C, 80 ± 5% relative humidity, and a L16:D8 photoperiod. 

Tomato plants (cv. ‘Marinice’) (De Ruiter Seeds, Bleiswijk, The Netherlands) were sown in trays. One week after germination, tomato seedlings were transplanted to 1 L pots with potting soil (Greenyard Horticulture, Ghent, Belgium). Plants were subsequently grown in the Greenlab research facilities of Biobest (16.5–22 °C and 70 ± 10% relative humidity).

### 2.2. Experiment 1: Pronematus ubiquitus Strain Comparison

In the first experiment, we assessed oviposition of two *P. ubiquitus* populations, collected in Belgium and The Netherlands. Each involved a single adult female *P. ubiquitus,* randomly selected from the mass-rearing units, which was then placed on a 4 cm^2^ tomato leaf section on top of moist cotton within a 5 cm Petri dish arena. These leaf discs had *Typha angustifolia* ad libitum as a food supplement once at the start of the experiment. Each female was allowed to lay eggs for four days. At this time, the total number of eggs laid by each female was counted. These arenas were kept at 23 ± 1 °C, 70 ± 10% relative humidity, and a L16:D8 photoperiod. Each treatment contained 12 replicates. The total oviposition was analyzed using generalized linear models (GLMs) with a Poisson error distribution [51]. In addition, contrasts among predators were determined with general linear hypothesis testing (function glht of the package lsmeans in R, Lenth 2016). 

### 2.3. Experiments 2 and 3: Oviposition of P. ubiquitus on Alternative Foods 

In these two experiments, the *P. ubiquitus* line collected in Belgium was studied for its oviposition ability by applying the same method as that of Experiment 1. For Experiment 2, the food sources offered for the mites were: clean tomato leaf, tomato leaf infected with powdery mildew, tomato leaf with tomato pollen, and tomato leaf with *T. angustifolia* pollen. Tomato pollen was obtained from flowers collected from tomato plants (cv. ‘Marinice’) (De Ruiter Seeds, The Netherlands) in the Greenlab facilities. Flowers were dried for 72 h at room temperature (20–23 °C) and relative humidity (30–50% RH). Thereafter, pollen could be removed easily from the flowers before being frozen at −18 °C. Experiment 3 had clean tomato leaf, tomato leaf with *T. angustifolia* pollen as per the previous experiment, and tomato leaf with the prey mites *T. fusarii* or *C. lactis.* For both prey mites, a mix of all life stages was offered. All food sources were offered ad libitum once at the start of the experiment. Each treatment in Experiment 2 had 12 replicates and Experiment 3 had ten replicates per treatment. The total oviposition of Experiment 2 was compared among treatments with a generalized linear model (GLM) with a Poisson error distribution [51]. For Experiment 3, the comparison among treatments was also performed with a GLM. However, the distribution for this case was a quasi-Poisson error distribution due to the overdispersion of the data. In addition, contrasts among predators were determined with general linear hypothesis testing (function glht of the package lsmeans in R [52]). 

### 2.4. Experiment 4: Frequency of Application of Typha Angustifolia for Pre-Establishing P. ubiquitus

For this experiment, a cage trial was performed on individual tomato plants cv. Merlice (De Ruiter Seeds, Bleiswijk, The Netherlands) to determine the ideal frequency of application of *T. angustifolia* pollen to promote *P. ubiquitus* (Belgium line) population growth. Cylindrical cages were used for this experiment having a diameter of 40 cm and height of 110 cm. At the start of the experiment the tomato plants had five to six leaves and height of 32 cm. Four frequencies of pollen application were evaluated in comparison with control without pollen supplement. Five treatments were arranged in a randomized complete block design with five replicates: *P. ubiquitus* without pollen supplement; *P. ubiquitus* + pollen supplement weekly; *P. ubiquitus* + pollen every two weeks; *P. ubiquitus* + pollen every three weeks; *P. ubiquitus* + pollen every four weeks. Fifty mixed stages of *P. ubiquitus* were introduced per plant at the first week of the experiment and an additional 100 one week later. A dose of 0.1 g of *T. angustifolia* pollen was evenly distributed on each plant with a soft hair brush over the entire plant at the frequency defined for each treatment. This amount of pollen per application corresponds to the recommended amount of 500 g/ha for supplementary feeding of phytoseiid predatory mites [40,41]. The first assessment was carried out two weeks after the first introduction of *P. ubiquitus*. Assessments involved counting the number of *P. ubiquitus* mobiles found on fifteen individual 7 cm^2^ leaflet sections per plant under a stereoscopic microscope. These leaflets were collected at random through the entire plant. These counts were repeated for five consecutive weeks following the second release. The number of mites was compared among treatments with a linear mixed-effects model (LME) with treatment and time as fixed factors and plant identity as a random factor to correct for repeated measures. Non-significant interactions and factors were removed from the model. In addition, contrasts among predators were determined with general linear hypothesis testing (function glht of the package lsmeans in R [52]). All analyses were performed using the statistical software R 3.6.1 [53]. 

## 3. Results

### 3.1. Experiment 1: Pronematus ubiquitus Strain Comparison 

No difference was observed in the reproduction capacity of *P. ubiquitus* collected in Belgium and The Netherland (Figure 1: GLM, χ^2^ = 0.34; *p* = 0.56). 

### 3.2. Experiment 2: Oviposition of P. ubiquitus on Naturally Occurring Food Sources on Tomato Crops

The different food sources significantly impacted the oviposition of *P. ubiquitus* (Figure 2: GLM, χ^2^ = 100.8; *p* < 0.001). The highest oviposition was recorded when *T. angustifolia* was offered as a food source, followed by tomato pollen (Figure 2). When mites had access to powdery mildew, only a marginal improvement in the oviposition was recorded (Figure 2). The lowest oviposition was observed when the tomato leaf had no additional food source (Figure 2).

### 3.3. Experiment 3: Oviposition of P. ubiquitus on Factitious Food Sources 

The oviposition of *P. ubiquitus* was influenced by the different food sources (Figure 3: GLM, F_2,39_ = 100.8; *p* < 0.001). Similar to the previous experiment, *T. angustifolia* induced the highest oviposition rate, followed by the tarsonemid prey mite *T. fusarii* (Figure 3). The prey mite *C. lactis* did not show any additional effect on the reproduction of *P. ubiquitus* compared to the tomato leaf alone (Figure 3). 

### 3.4. Experiment 4: Frequency of Application of T. angustifolia for Pre-Establishing P. ubiquitus

There was a significant interaction between the frequency of application of *T. angustifolia* and time on the population of *P. ubiquitus* (Figure 4: LME, χ = 102.9, *p* < 0.001). Weekly application of pollen resulted in the highest number of *P. ubiquitus,* followed by applications every other week (Figure 4). However, the application of pollen every three or four weeks did provide an initial benefit; this benefit was not sustained during the entire trial (Figure 4). 

## 4. Discussion

In this study, natural and factitious food sources were evaluated for the predatory mite *P. ubiquitus*. For phytoseiid predatory mites, differences in life history traits have occasionally been observed between different strains of the same species [54,55]. Here, we did not observe differences in egg laying on *T. angustifolia* pollen between the two lines of *P. ubiquitus* collected from Belgium and The Netherlands. This result is to be expected because both collection sites were close and no morphological differences were observed between strains (de Vis and Ueckermann, personal communication)**.** We opted to undertake the subsequent trials using only the Belgium strain because we had worked with this strain longer and did not have any reason to believe that the strains would behave differently on the other food sources. 

Overall, *Typha angustifolia* pollen was found to be the most suitable food source for reproduction compared to a diet consisting of fungus (i.e., powdery mildew) or prey mites. These results confirm the findings of previous studies showing pollen to be an excellent food for Iolinidae predatory mites, such as *P. ubiquitus* and the closely related *Homeopronematus anconai* (Baker) (Acari: Iolinidae) [35,37]. However, tomato pollen was found to be inferior as compared to *T. angustifolia* pollen. Only a few predatory mites were observed on tomato plants when tomato pollen was the only food source (i.e., no *T. angustifolia* pollen was added). This confirms that pollens derived from different plant species can differ greatly in their nutritional quality for predatory mites [6,56]. Furthermore, other pollen features such as grain size, structure, and exine thickness determine the suitability of a pollen food source [57]. However, both pollen species tested here have a diameter of ca. 20 µm [57,58]. Therefore, size cannot explain the observed differences in reproductive output.

Although tomato is a self-pollinating plant, the agitation of the anthers (e.g., by pollinators or by wind) facilitates shedding of the pollen grains. As a result, tomato pollen tends to be confined in flowers and only a limited amount of pollen will drop down to the leaves below to serve as food for the mites. Tomato growers introduce bumblebee hives in the greenhouse to facilitate crop pollination. By engaging in so-called “buzz-pollination”, bumblebees promote the release of tomato pollen from the anthers [59,60], resulting in optimal fruit set and yield [58]. In this study, no bumblebees were present during the trial as plants were confined individually in cages. It remains to be investigated whether bumblebee presence would allow pollen to drop down more due to the buzz-pollination, thereby supporting populations of *P. ubiquitus* on the tomato crop. Finally, pollen quality and quantity depend on the tomato cultivar [61,62], which has been shown to influence bumblebee behavior in tomato crops [63]. Similarly, the reproductive and population growth performance of *P. ubiquitus* may be affected by the tomato cultivar itself via plant feeding.

Tydeid mites are well known to engage in fungivory. For example, *Orthotydeus lambi* (Baker) was found to reduce the incidence and severity of grape powdery mildew, *Uncinula necator* (Schwein.) Burrill on both potted and field-grown vine plants [64,65]. Recently, Pijnakker et al. [3] reported the potential of *P. ubiquitus* to control powdery mildew on tomato plants. The reproductive output of *P. ubiquitus* on powdery mildew in our laboratory trial was lower than that on a pollen diet and similar to that of the tomato leaf-only treatment. Similarly, Hessein and Perring [37] reported that the presence of the fungus *Cladosporium cladosporioides* (Fres.) de Vries on grape leaf arenas yielded the same low number of *H. anconai* as compared to a grape leaf-only treatment. In contrast, high numbers of mites were retrieved when grape leaf arenas were supplemented with *Typha* sp. pollen. Although feeding on powdery mildew conidia and mycelia was observed, the exact mechanisms through which *P. ubiquitus* controls powdery mildew on tomato, as reported in the study of Pijnakker et al. [3], remains to be elucidated. Being a plant feeding mite, indirect plant-mediated effects might be triggered by feeding on the plant tissue, similar to the observations in other plant-feeding natural enemies [66]. Further study is required to investigate whether the biocontrol effect of *P. ubiquitus* on powdery mildew is due to feeding on the fungus, plant-mediated effects, or a combination thereof.

Food supplementation has become a standard practice in biocontrol programs in several crops to support the pre-establishment of generalist predators (Messelink et al. 2014; Pijnakker et al. 2021; Benson and Labbé 2021). Astigmatid prey mites are a relatively cheap alternative food source explored as a food supplement on crops, mainly to feed phytoseiid predatory mites [2]. The reproduction of *P. ubiquitus* on *C. lactis* in our study was negligible. Hence, *C. lactis* will most likely not be suitable for supporting *Pronematus* populations in a tomato crop. By comparison, the tarsonemid prey mite, *T. fusarii*, allowed reproduction in *P. ubiquitus*, albeit to a lesser extent than *T. angustifolia* pollen. Body size is an important factor determining the outcome of prey–predator interactions [67,68] and may explain why smaller tarsonemid prey mites would be more suitable than the larger *C. lactis*. *Typha angustifolia* pollen was clearly superior to the other foods tested, and although pollen would be the preferred food supplement based on our laboratory trials and previous results, it may nevertheless be interesting to test combinations of different food sources for rearing or maintaining on the crop. Mixing two diets was reported to yield higher reproduction than offering single diets in predatory mites [69,70,71].

Based on the results of the laboratory trials, we tested the impact of *T. angustifolia* pollen application frequency on *P. ubiquitus* establishment on tomato plants. Our results clearly show that pollen needs to be supplemented at least every other week to allow a good population build-up of *P. ubiquitus*. Populations can be maintained using less frequent applications (every three or four weeks). For the phytoseiid predatory mite *Iphiseius degenerans* Berlese on cucumber, van Rijn et al. [40] found that 14-day-old *Typha latifolia* recollected from the crop still allowed immature development and reproduction, although at a lower rate compared to freshly applied pollen.

## 5. Conclusions

In summary, we showed that *T. angustifolia* pollen is a good food source to sustain the reproduction of *P. ubiquitus*. Application of *T. angustifolia* pollen allows the build-up of *P. ubiquitus* populations on tomato plants when applied at a frequency of at least once every two weeks at a dose rate of 500 g of pollen per hectare. Tomato pollen was found to support the reproduction of *P. ubiquitus*, but to a far lesser extent than *Typha* pollen. Finally, our laboratory trials indicated tarsonemid prey mites, such as *T. fusarii,* show potential to support populations of *P. ubiquitus*, whereas the larger astigmatid prey mite *C. lactis* was not suitable.

## Figures and Tables

**Figure 1 insects-12-01111-f001:**
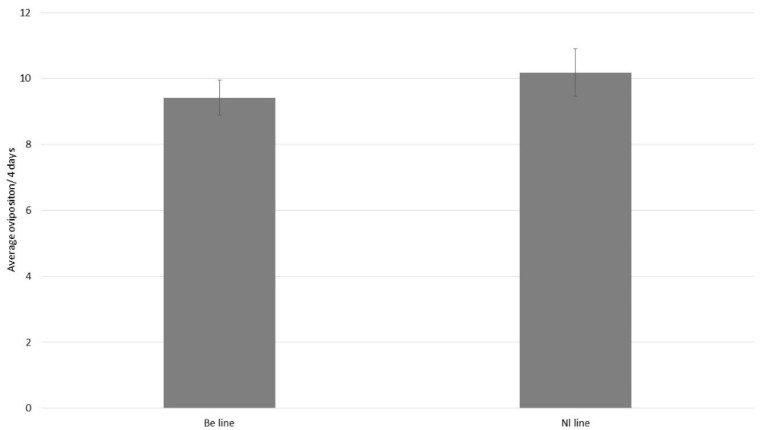
Average of the total oviposition for four days (mean ± SE) for *P. ubiquitus* either collected in Belgium (Be line) or The Netherlands (Nl line) on a tomato leaf with the addition of *T. angustifolia* pollen. There were no significant differences between the two lines (GLM, χ^2^ = 0.34; *p* = 0.56).

**Figure 2 insects-12-01111-f002:**
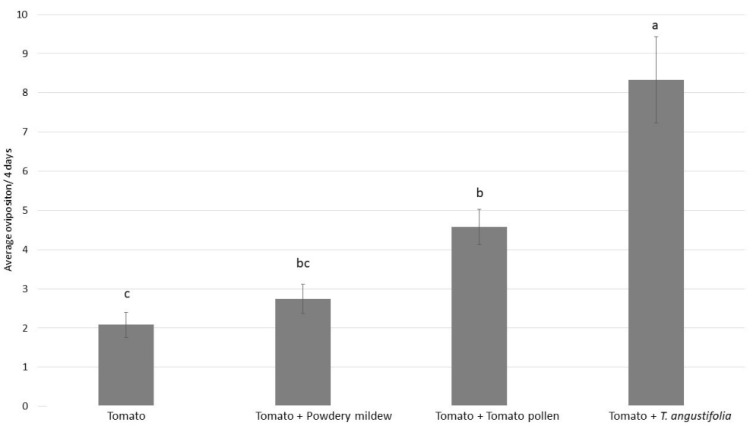
Average of the total oviposition over the course of 4 days (mean ± SE) for *P. ubiquitus* on tomato alone, tomato with powdery mildew, tomato with tomato pollen, and tomato and *T. angustifolia.* Different letters above the bars denote significant difference among treatments (contrast after GLM, *p* < 0.05).

**Figure 3 insects-12-01111-f003:**
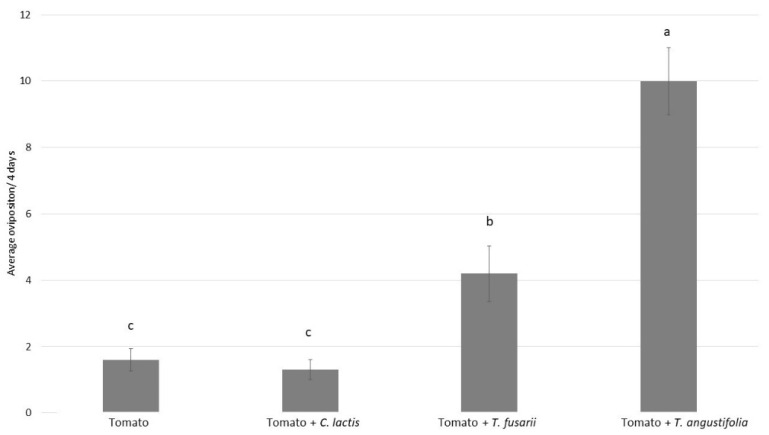
Average total oviposition over four days (mean ± SE) for *P. ubiquitus* on tomato alone, tomato with *C. lactis*, tomato with *T. fusarii* and tomato with *T. angustifolia.* Different letters above indicate differences among treatments.

**Figure 4 insects-12-01111-f004:**
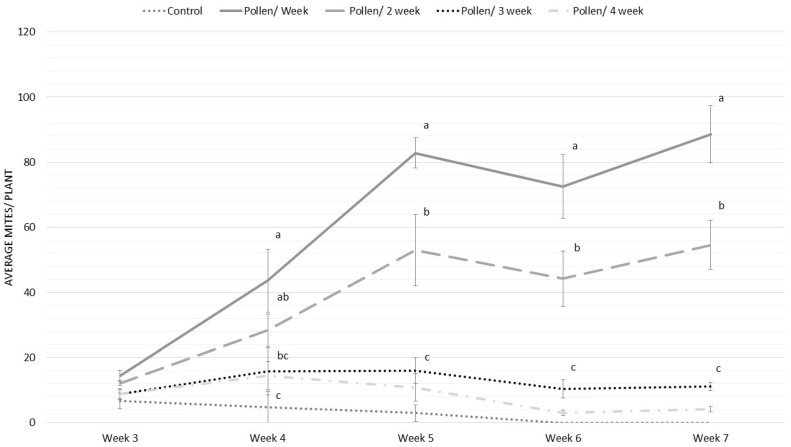
Population dynamics of *P. ubiquitus* (mean ± SE) when fed with a different *T. angustifolia* frequency. Control plant did not receive pollen. Pollen/week represents weekly pollen addition, Pollen/2 week means pollen addition every other week, Pollen/3 week means pollen application every 3 weeks, and Pollen/4 week means one pollen application every four weeks. Different letters above the data points denote significant differences among treatments during that week (contrast after LME, *p* < 0.05).

## Data Availability

The raw data are available from the corresponding author upon request.

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
