# Peer review of "Evaluation of Natural and Factitious Food Sources for Pronematus ubiquitus on Tomato Plants"

_insects, 2021, doi:10.3390/insects12121111_

Round 1

Reviewer 1 Report

I read the manuscript entitled “Evaluation of natural and factitious food sources for Pronematus ubiquitus on tomato plants” which was submitted by Marcus V. A. Duarte et al. to Insects Journal. The subject of this study is very important for including knowledge about the nutritional value of various alternative food sources that the mite Pronematus ubiquitus could encounter or be applied to the tomato crop. The paper is in very good shape with respect to the language and presentation. The manuscript can be accepted after minor revision for publication at the Insects Journal. I do not need to re-examine after corrections. Some suggestions are at below:

Line 28: Typha angustifolia should be written in italics.

Line 68- 71: The "secondly" on line 68 can be confused with the "secondly" on line 71. It should be revised.

Line 78: P. ubiquitus should be written in italics.

Line 106-107: How much Typha angustifolia pollen was given as food? Was the required pollen given on the first day for a total of four days? Or is a certain amount given each day? How many replications were done in the experiment?

Line 114-130: What were the numbers of replication for experiment 2 and 3? Pollen amount? Prey amount?

Line 132: What was the size of the cage? What was the length of the tomato plants used in the experiments? How many leaves did the plants have?

Author Response

Dear Reviewer,

Thank you for taking the time to review and provide insightful comments for our manuscript. Bellow the responses to your comments.

Line 28: Typha angustifolia should be written in italics.

R: Done.

Line 68- 71: The "secondly" on line 68 can be confused with the "secondly" on line 71. It should be revised.

R: The secondly from line 71 was substituted for "The"

Line 78: P. ubiquitus should be written in italics.

R: Done.

Line 106-107: How much Typha angustifolia pollen was given as food? Was the required pollen given on the first day for a total of four days? Or is a certain amount given each day? How many replications were done in the experiment?

R: Detailed the amount and feeding frequency. Number of replicates were added.

Line 114-130: What were the numbers of replication for experiment 2 and 3? Pollen amount? Prey amount?

R: Also detailed the amount and replicates.

Line 132: What was the size of the cage? What was the length of the tomato plants used in the experiments? How many leaves did the plants have?

R: Added the details.

Best regards,

Marcus

Reviewer 2 Report

This manuscript "Evaluation of natural and factitious food sources for Pronematus ubiquitus on tomato plants" evaluates alternative food sources for a new iolinid predatory mite which can feed on other mites, pollen and fungal spores. Previous research has established the potential for this species to manage russet mites and powdery mildew on plants, including tomato. The assays conducted here identified additional food supplements that can help establish or maintain populations of P. ubiquitus in the absence of prey. There was some oviposition observed on tomato leaves with tomato pollen and T. fusarii, but the best supplement was Typha pollen. Further greenhouse assays observed establishment of P. ubiquitous at 2-week interval applications of Typha.  

The experimental design and analysis for this paper is appropriate and the paper is well written overall. Suggestions to improve grammar and clarity are made in the pdf.

The only major suggestion I have is to re-order the discussion to explain why the Belgium strain was assessed and not the Netherlands strain. Without an explanation, the first experiment seems unnecessary. The reader is left wondering why the same experiments were not conducted for both to see if there were any differences. It may be more appropriate to put that test into supplementary data and to state in the methods that this strain was selected due to prior work. 

I also put a note in the statistics section to clarify the difference between general and generalized linear models. The authors refer to both in the methods section but I believe they are referring to generalized model which does not require continuous variables or normal distribution.  

I believe after these corrections are made the paper will be ready for publication. 

Author Response

Dear Reviewer,

Thank you for taking the time to review and provide insightful comments for our manuscript. 

The reason why we opted to continue with only the Belgium strains is due to the results of our first experiment which were done with the best food source we expected to be the best (T. angustifolia pollen) for both strains, we opted to do the subsequent trial with only this strain which we have been working with longer. The strains could indeed have behaved differently with the other food sources, but we found it redundant to continue all the trials with both strains. We've added these reasons to the begining of the discussion as suggested.

We indeed meant generalized linear models and have corrected it in the text. We have also adapted the text with the comments in the PDF.

Best regards,

Marcus

Reviewer 3 Report

Overall comments

This study provides important information regarding the effect of diet on reproductive rate of a predatory mite that can contribute to control of arthropod pests on tomato crops. The manuscript is very well written and clearly structured.

Specific comments

Line 23, (and Introduction, line 78) italicise “P. ubiquitous”

Line 28 italicise “Typha angustifolia”

Need to mention that in both laboratory experiments included Typha angustifolia pollen.

Introduction

Line 58 insert ‘on’; “…omnivorous mite that feeds on a variety of plant-derived…”

Materials and Methods

How many replicates of single adult females were there for each of the ‘treatments (including controls or different populations) in each of the 3 laboratory experiments?

Line 122 Do you mean you used the exact same leaf in Experiment 3 that you used in Experiment 2, as that is how the sentence currently reads. Or do you mean “In experiment 3 we used the same leaf segment method and included the controls, clean tomato leaf and tomato leaf with T. angustifolia pollen in addition to the two prey mite species T. fusarii or C. lactis.”

Line 145 Was that a single 7cm2 leaf segment per cage? If only one leaf segment per cage then please say so.  Or did you count mites on more than one leaf segment? If more than one leaf segment then how many per cage?

Results

In the Figure captions please explain what the error bars represent; e.g. LSD or confidence limits etc.?

Discussion

Line 02 – 205 Sentence starting with “Moreover…” is a bit confusing. Suggest that the authors’ rewrite; “Only a few predatory mites were observed on tomato plants when tomato pollen was the only food source (i.e. no T. angustifolia pollen was added). This confirms….”

Author Response

Dear Reviewer,

Thank you for taking the time to review and provide insightful comments for our manuscript. Bellow the responses to your comments.

Line 23, (and Introduction, line 78) italicise “P. ubiquitous”

R:  Done

Line 28 italicise “Typha angustifolia”

R: Done.

Need to mention that in both laboratory experiments included Typha angustifolia pollen.

R: Mentioned in line 107 and 124.

Line 122 Do you mean you used the exact same leaf in Experiment 3 that you used in Experiment 2, as that is how the sentence currently reads. Or do you mean “In experiment 3 we used the same leaf segment method and included the controls, clean tomato leaf and tomato leaf with T. angustifolia pollen in addition to the two prey mite species T. fusarii or C. lactis.”

R:  Changed it to: "Experiment 3 had the same treatments with tomato leaf and tomato with T. angustifolia..."

Line 145 Was that a single 7cm2 leaf segment per cage? If only one leaf segment per cage then please say so.  Or did you count mites on more than one leaf segment? If more than one leaf segment then how many per cage?

R: Added "one" to indicate the number of areas counted.

In the Figure captions please explain what the error bars represent; e.g. LSD or confidence limits etc.?

R: Added (mean ± SE) indicating what the bars represent.

Line 02 – 205 Sentence starting with “Moreover…” is a bit confusing. Suggest that the authors’ rewrite; “Only a few predatory mites were observed on tomato plants when tomato pollen was the only food source (i.e. no T. angustifolia pollen was added). This confirms….”

R: Done.

Best regards,

Marcus